



# Impact data bases application for natural and technological risk management

Nina I. Frolova[1], Valery I. Larionov[1], Jean Bonnin[2], Sergey P. Suchshev[3], Alexander Ugarov[3], Nataliya Malaeva[3]

[1]Seismological Center of IGE, Russian Academy of Sciences, Moscow 101000, Russia
[2] Institut de Physique du Globe, University of Strasbourg, Strasbourg F-67084, France
[3] Extreme Situations Research Center, Moscow 127015, Russia

*Correspondence to*: Nina I. Frolova (frolova@esrc.ru)

**Abstract.** Impact databases development and application for risk analysis and management promotes the usage of self-learning computer systems with elements of artificial intelligence. Such systems learning could be successful when the databases store the complete information about each event, parameters of the simulation models, the range of its application and residual errors. Each new description included in the database could increase the reliability of the results obtained with application of simulation models. The calibration of mathematical models is the first step to self-learning of automated systems. The article describes the events' database structure, and examples of calibrated computer models as applied to the impact of expected emergencies and risk indicators assessment. Examples of database statistics usage in order to rank the subjects of the Russian Federation by the frequency of emergencies of different character, as well as risk indicators are given.

## 1 Introduction

Analysis of the natural and technological emergencies consequences gives an evidence that natural hazards and technological disasters pose an increasing threat to the safety of citizens and the economy of the Russian Federation. The increasing severity of the impact indicates the need to improve the effectiveness of measures aimed at risk reduction. The Ministry of Emergency Situations (EMERCOM) of the Russian Federation considers preventive measures as the priority. They are based on application of information systems (IS) that provide reliable forecasts, including a reliable assessment of a spatially distributed indicator that characterizes the safety of a region, which makes it possible to rationally distribute forces and resources in order to reduce risk.

Another important task, which may be solved with IS usage, is to enhance the efficiency of rescue operations. This could be achieved by higher reliability of the operational forecast of situations based on the data contained in the database of events (DB). Examples of successful rescue operations accumulated in DB facilitate the decision-making process and reduce the time when people stay in the affected area, which results in decreasing the fatality likelihood.



The article focuses on increasing the reliability of expected emergency impact through calibration of IS mathematical models.

The considered event DB (Kachanov et al., 2011; Kachanov et al., 2014) is interdepartmental and since 1992 it has been maintained by the resources of the National Center for Crisis Management EMERCOM of Russia. Information about events and their impact comes from the EMERCOM departments and different agencies, including the system 112. Currently, the
DB includes more than 30,000 records with a detailed description of emergency situations, rescue operations, and an assessment of their effectiveness.

For seismic events that occur quite rarely in the Russian Federation, the list has been expanded retrospectively with records of earlier years from other sources. Archival data are used, the relevance of which is supported by the efforts of researchers at the Russian Academy of Sciences (RAS) and various international organizations. First of all, the following DB are used as
the "Extremum" DB, the "Strong and Felt Earthquakes of the Northern Eurasia" DB, the "EM DAT" DB of the Brussels Center for Epidemiology of Disasters, as well as the DB of SwissRE and MunichRe reinsurance companies. Important sources of information also include the "CATNAT" database, which is used to calibrate the "PAGER" IS (Allen et al., 2009) and the Cambridge Earthquake Impact Database (www.ceqid.org) (So, 2014; Spence et al., 2011), used for calibration of vulnerability functions for buildings and structures of various types when calculating seismic risk. Extensive macroseismic
information is of great importance as well. It is available as reconnaissance team reports, scientific papers, Earthquake Atlases of individual countries (Kalmetieva et al., 2009; Babayan, 2006), and in the form of electronic DBs (Rovida et al., 2016; Varazanashvili et al., 2018).

The article gives examples of calibrated models as applied in near real time earthquake loss estimation practiced at the EMERCOM of Russia. The results of EMERCOM DB statistical data analysis for visualization of statistical information in
the Russian Federation with different levels of emergency frequency and various values of risk indicators are also listed.

## 2 Structure of the EMERCOM database of events

The EMERCOM database about events title partially describes its content, namely, data "about events", and accurately corresponds to the structure of the records. It allows retrieval of the time and location of an event; operations carried out and their effectiveness; used forces and resources, their sufficiency and estimation of the operations, as well as the reported
social and economic losses.

Each DB entry is, to some extent, an event passport which is characterized as an emergency situation. A unique number is assigned to DB entries, which allows connection to entries in the databases belonging to different agencies interested in sharing and obtaining the information in order to normalize the emergency situation. Due to the uniqueness of entry number and the event standard description corresponding to the regulatory documentation, event entry can become part of the state
or international spatial data infrastructure (https://doi.org/10.5281/zenodo.2682654).



The DB is part of the EMERCOM IS data storage developed and maintained up to date within of the Ministry information program. Besides the data storage, the EMERCOM IS includes software consisting of a set of blocks and interfaces. The system was developed using modern Web-technologies (Izmalkov, 2017a; Izmalkov, 2017b).

The structure of the DB about events, which is part of the EMERCOM IS data storage primarily aim at providing

management automation and solving the following tasks with the system application:

– inventory of hazards sources and elements at risk;

– forecast of the level of hazard and preparation of data for warning, including the information for mass media;

– monitoring and recording of events characterized as an emergency;

– analytical decision support in case of emergencies, including the method of mathematical modeling of hazardous

impacts, the response of elements at risk and the forecast of losses;

– analytical support for risk assessment and mapping;

– drafting reports on the situation, weather conditions, needed forces and resources, as well as the assessment of possible losses.

A scheme of the EMERCOM IS is presented in the figure (fig. 1), where the enlarged functional units, the data storage, and

the user interface are highlighted. The names of units reflect their functionality, which is supplemented by a description of the input and output information. The structure identifies the main units of IS, including software tools under the titles "Inventory", "Analytics", "Operational Management".

The "Inventory" block is assigned to collect and process the data necessary for creating documents called the "territory passport", which contains:

• name and affiliation of objects (territories);

• location of the facilities subject to inventory;

• parameters of hazard sources;

• vulnerability functions of elements at risk;

• availability, deployment, as well as readiness indicators of forces and resources needed for preventive measures

implementation and rescue operations.

WEB-interface resources accessible by local (as close as possible to the source of information) employees of the Unified state system of emergency prevention and response (RSES) are used to collect data. As an output, a "Passport of the Territory" is developed, which contains information in order to estimate matching of the hazard level, the measures taken and the risk indices.

The "Operational Management" block deals with recording, qualification, and operational management of forces and resources during events characterized as an emergency. The output of this block is an "Event Card". It includes descriptions of the emergency impact, which can be updated over time, and a set of documents ensuring the management of forces and resources during rescue and other urgent actions.



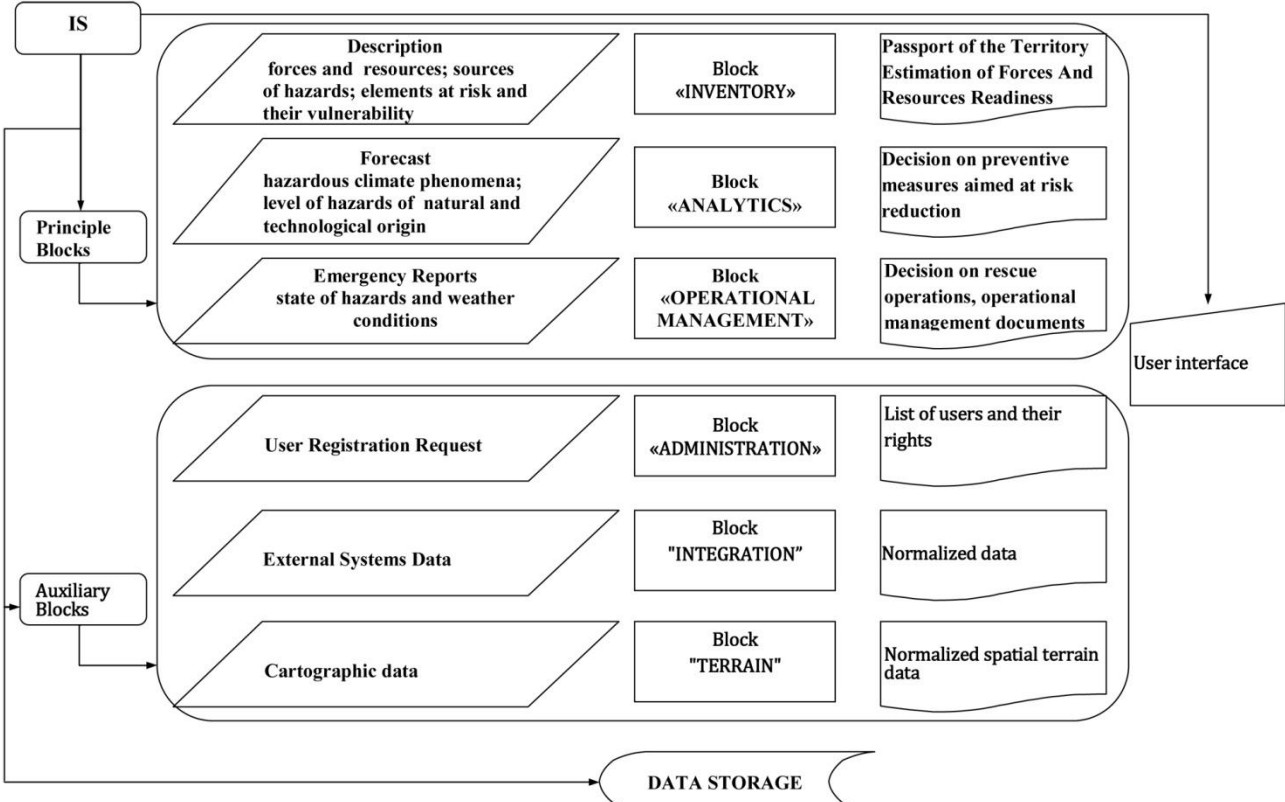

**Figure 1: Structure of EMERCOM Information System.**

The "Analytics" block contains mathematical models of the main types of hazardous processes and their impact, which allow a possible situation in the affected zone to be predicted. The models meet special requirements due to high demands on the speed of response to input data and ensuring the possible minimum error in results of forecast which may depend on input data measurement and transmission.

The "Analytics" block mathematical models are calibrated for the site of an event in order to ensure acceptable accuracy of
simulation results. Zones with a set of calibration parameters of mathematical models valid for each area are visualized on the map. Calibration, in this case, largely repeats the learning process of artificial intelligence systems (AI). After calibration, any mathematical model implemented as a calculation software module, in the first step, determines the zone of its calibration and retrieves the calibration parameters assigned to this zone. After that, the computation is done.

All calibration parameters associated with the territories and design modules are included in the EMERCOM IS data storage.
The figure (fig. 2) shows the data storage structure and the place of calibration data tables in this structure.

In Figure 2, bold lines highlight the "Database with a description of events" and "Zones with stable calibration parameters of mathematical models". Zones are based on the collected event data in the process of calibrating mathematical models. Each zone relates to its mathematical model and set of calibration parameters.



It should be noted that the description of seismic events has some features important for understanding the examples of

calibrated mathematical model application for estimating the possible shaking intensities.

Figure 3 shows the features of the structure of tables with a description of seismic events. Structured description of a seismic event may be accompanied by various materials such as maps; tables; photos and other documents. When calibrating the mathematical models applied for simulation of expected shaking intensities, the tables of "Earthquake Parameters" and "Data on Macroseismic Survey" are used during the calibration.


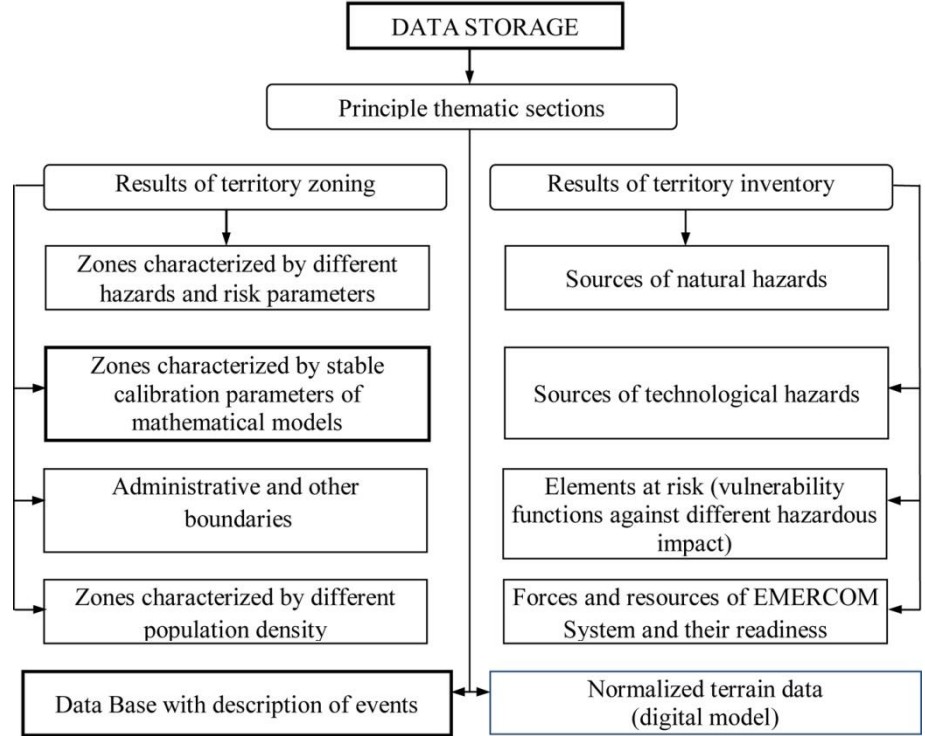

**Figure 2: Structure of EMERCOM Information System data storage.**

In the case of the Kultuk earthquake of 2008, which was felt over a wide area in the Russian Federation and Mongolia, and the damage to buildings was recorded in settlements located 10-30 km away from the epicenter, the following field observations and their analysis complement the "Data on Macroseismic Survey" table:

•       a map of the observed macroseismic effect, where the first two isoseismals show the area of ground shaking of 7–8 and 7 grades on the MMSK-86 intensity scale (Shebalin et al., 1986);

•       photos of damage to buildings of different types;

•       summary tables of macroseismic effect in settlements based on various data, including the behavior of buildings and people, as well as indoor objects.

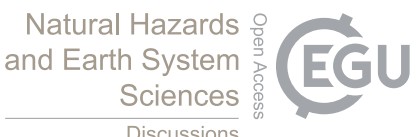

As a result of the Kurchaloy earthquake of 2008, 13 people died, the observed macroseismic effect was also 7-8 grades of the MMSK-86 seismic intensity scale. The earthquake was felt over a wide territory, including the Chechen Republic, the Ingush Republic, the Republic of Dagestan, the Republic of North Ossetia-Alania, Kabardino-Balkaria, the Stavropol Territory, the southern regions of Kalmykia, Georgia, the northern regions of Azerbaijan and Armenia. The greatest destruction occurred in the Kurchaloy, Gudermes, Shalinsky and Nozhay-Yurt districts of the Chechen Republic. The most

severely affected were the settlements of Kurchaloy and Mayrtup. The entries in the database tables about this event include additional materials on observed macroseismic data and results on engineering analysis of the impact:

• the isoseismal map of the Kurchaloy earthquake of October 11, 2008 with seismic intensity $I$ from 7–8 to 4–5 grades of MMSK-86 scale;

• revised instrumental and macroseismic parameters of the earthquake rupture zone;

• photos and descriptions of damage to residential buildings, healthcare facilities, schools and preschool institutions, cultural and sports facilities, mosques and administrative buildings;

• information about response measures, including data on medical and other assistance to the affected population.

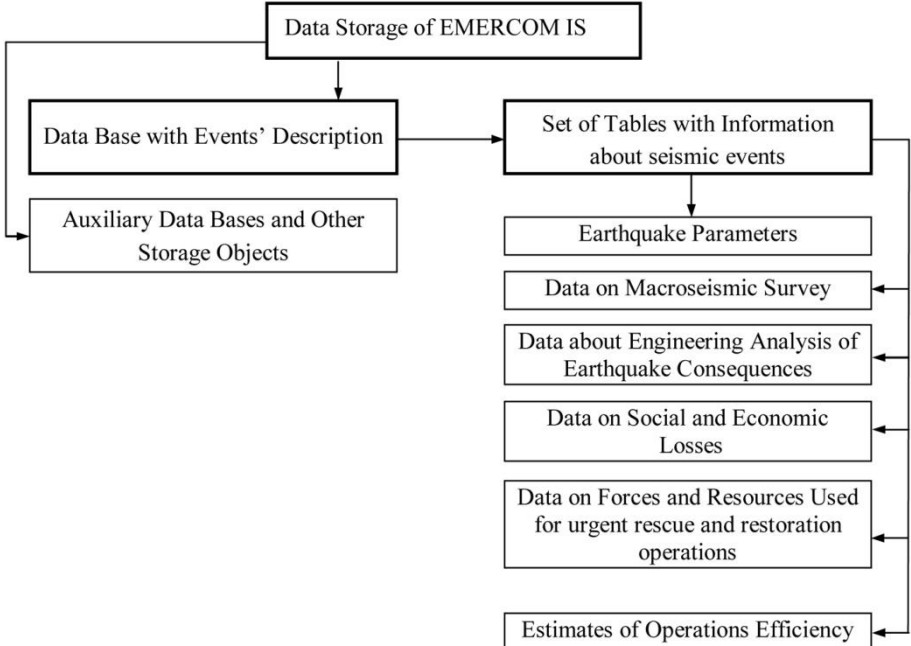

**Figure 3: Data Base on events tables with earthquakes description.**

Calibration means fitting suitable parameters of the mathematical model assigned for simulating the macroseismic field and

identification the boundary of this zone within which averaging is permissible. Calibrated models should provide acceptable accuracy of estimating the situation parameters for all events in the zone. The calibration parameters of macroseismic field



model include: regional intensity attenuation coefficients in the Shebalin equation (Shebalin, 1977); the orientation of the axes of macroseismic field ellipse; the ratio $k$ of the major and minor axes of macroseismic field ellipse (compression ratio). Next section shows the application of EMERCOM DB with calibrated parameters of a computational module in order to

simulate the shaking intensity distribution in near real time with acceptable accuracy and to get reliable forecast of the whole situation in the stricken area.

### 3 Examples of calibrated models application for earthquake loss assessment

This section gives an example of calibrated macroseismic field models usage to simulate earthquake losses in emergency mode. We investigated the North Caucasus, which is one of the seismically active regions of the Russian Federation

characterized by a high population density. Here, strong and felt earthquakes occurred quite frequently. Over the past 100 years, more than 20 earthquakes took place in this area with shaking intensity $I_0 \geq 6$: in 1976, the Chernogirskoe earthquake with $M=6.4$, $I_0=8$–9; Terskoye (Eldarovskoe) earthquake on August 10, 1912 with $M=5.7$ and $I_0=8$; Vedenoskoe earthquake on October 24, 1933 with $M=5.2$, $I_0=7$–8; several earthquakes with $I_0=7$: Dagestanskoe-I on February 23, 1785 with $M=5.5$; Argunskoe-I on May 10, 1928 with $M=4.7$; Argunskoe-II on March 2, 1966 with $M=4.9$; Achkhoy-Martanovskoe earthquake

on June 17, 1969 with $M=5.1$; Starogroznenskoe earthquake on May 26, 1971 with $M=4.1$; Salatausskoe earthquake on December 23, 1974 with $M=5.0$. The data on past earthquakes impact allowed us to calibrate the seismic intensity attenuation models and the vulnerability function of residential buildings. It was possible to identify the zone boundaries with stable parameters of the macroseismic field (Frolova and Ugarov, 2018). The zone boundaries are shown by bolt blue rectangle (fig. 4) on the map of the epicenters of the strongest earthquakes in the central part of the Terek-Caspian trough

against the background of the main structural elements according to the "Scheme of seismogenic structures of the Chechen Republic and adjacent territories". The authors came to the conclusion about the expediency of applying the Shebalin equation for Dagestan ($b=1,5$; $v=3,6$; $c=3,1$), compression ratio $k = 1.5$ and elliptic isoseismal orientation at an angle of 54° (Frolova and Gabsatarova, 2015) for near real time earthquakes loss assessment in the designated area.

Figure 5 shows an example of evaluating the consequences of the 2008 Kurchaloy earthquake which occurred in a

designated area, using calibrated Extremum IS models. The number of expected fatalities in emergency mode at $P = 0.9$ ranged from 12 to 100 people, which corresponded to the reported data on the death of 13 people during this earthquake. The error in determining shaking intensity $I$ when using calibrated models does not exceed 30%, unlike similar evaluations of PAGER System, where discrepancies in several settlements were 1-2 grades of intensity scale.

Based on the macroseismic survey data from past events that have occurred in the territory of the Russian Federation and the

CIS countries, the authors obtained calibration parameters of the macroseismic field models for other earthquake prone areas. Given different availability of macroseismic data for studied regions of the Russian Federation and neighboring countries and their heterogeneity, not all quasi-stable parameters characterizing the attenuation of seismic intensity $I$ can be determined for identified zones. Thus, for the Sochi zone, the calibration parameters of the Shebalin equation were $b=1.48$;
v= 4.0; *c*=2.73, and the compression ratio *k* = 2. For the Anapa zone, the tendency of elliptic isoseismal orientation towards

the anti-Caucasian direction was detected, which made it possible to increase the reliability of near real time loss estimates, including those after the Nizhnekubanskoe earthquake of November 9, 2002.

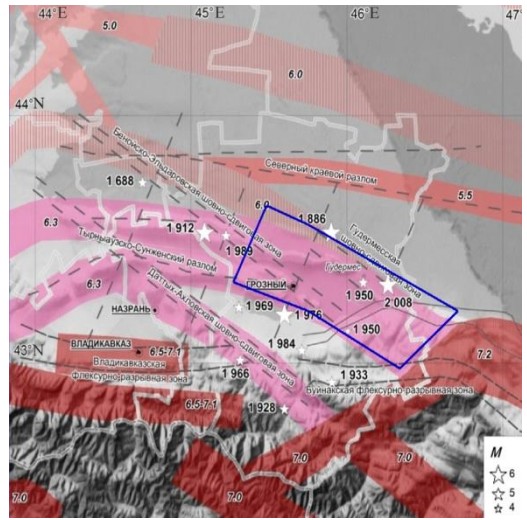

**Figure 4: The zone with stable parameters of macroseismic field; map background according to (Nesmeyanov et al., 1996; Rogozhin et al., 2013).**

**Figure 5: Calibrated model application for simulation the 2008 Kurchaloy earthquake consequences.**

Despite the limited material, work on the calibration of the macroseismic field models continues, taking into account that in

the future, when obtaining additional data, the boundaries of zones with quasi-stable parameters and the values of the parameters can be refined. The work (Frolova et al., 2019) shows the results of joint research with employees of the Federal Geophysical Survey RAS (FGS RAS) on the identification of a zone with quasi-stable parameters of the macroseismic field (fig. 6), which can be used later for rapid calculations of damage and losses in emergency mode.



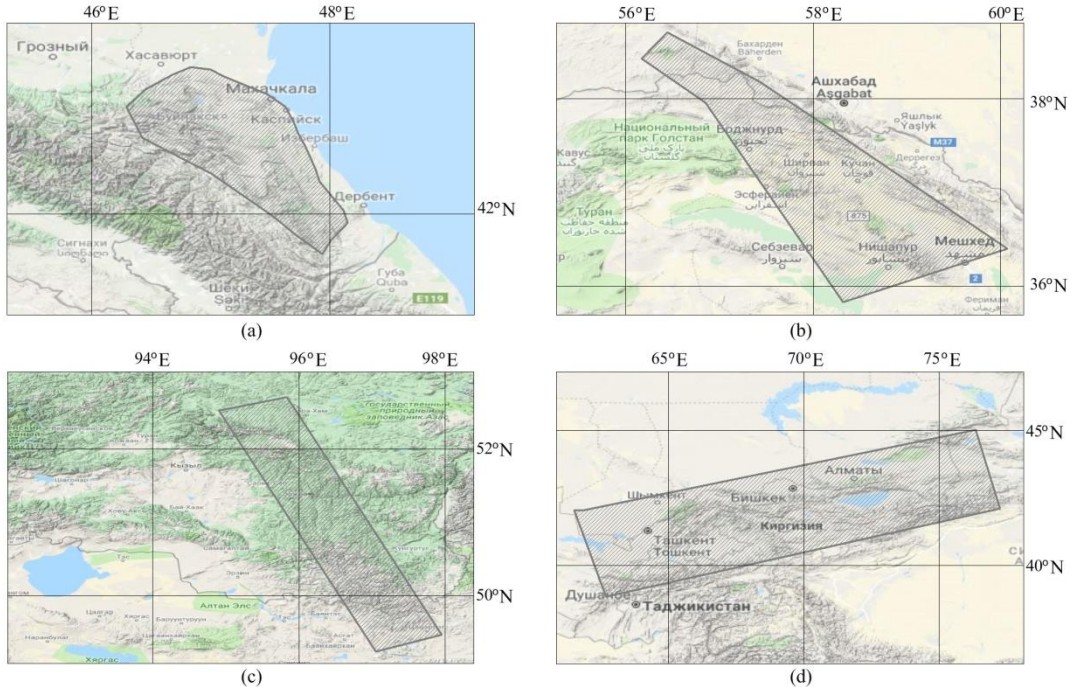

**Figure 6: Zones with quasi stable parameters of macroseismic field: a – in the Caucasus; b– in the Kopetdag-Caspian region; c – in the Altai Sayan region; d– in Central Asia by (Frolova et al., 2019).**

**4 Results of the EMERCOM DB on events usage for statistical analysis of emergencies in risk reduction**

The EMERCOM DB on events for the period from 1992 to 2018 was used not only to calibrate loss assessment models to improve the reliability of near real time estimates, but also for statistical assessments of emergencies to visually display the situation with emergencies in the country, as well as to develop maps of risk indicators.

Thus, our analysis of emergencies for the period under consideration provided the distribution of the frequency of occurrence of technological, natural and biological-social emergencies. The percentage of technological emergencies in the territory of

the Russian Federation is more than 70% of the total number of emergencies (fig. 7). Natural hazards, due to their intensity and duration, can have a negative impact, exceeding the scale of technological emergencies. The percentage of economic loss caused by natural emergencies is more than 80% of the emergency total loss in the Russian Federation per year (fig. 8).

According to the analyzed statistical data for the period from 1992 to 2018, more than 30% of the subjects of the Russian Federation are characterized by the frequency of annual emergencies exceeding the average level (more than 16 emergency

situations per year).

More than 50% of emergency situations occurred in the Central (18%), Volga (17.4%) and Siberian (18.4%) federal districts with a population of more than 60% of the total population of the Russian Federation. The gross regional product of the Central, Volga and Siberian federal districts is more than 50% of the total gross domestic product of the Russian Federation.


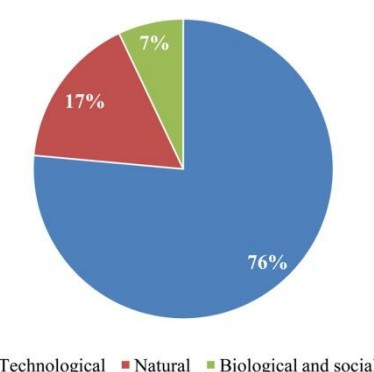

**Figure 7: Frequency of different types' emergency.**

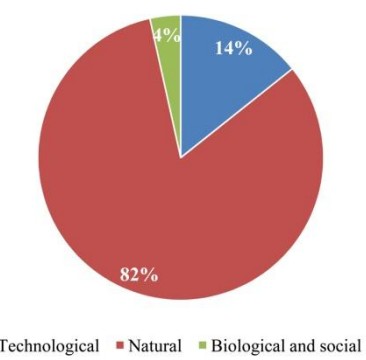

**Figure 8: Economic losses due to emergencies.**

Among the EMERCOM DB data on technological emergencies, explosions and fires at sites whose affected areas extend beyond their territories are especially dangerous. Their impact results in destruction of buildings, other facilities and structures, as well as in civilian casualties. Emergencies caused by fires account for more than 50% of the total number of technological emergencies (fig. 10).

The frequency of emergencies occurrence during transport accidents, including rail, road, sea and pipeline transport, is about 30% of the total number of technological emergencies.

More than 50% of technological emergencies occurred in the Central (21%), Volga (19.2%) and Siberian (16.1%) federal districts with a population exceeding 60% of the total population of the Russian Federation (fig. 11). The rate of occurrence of such emergencies varies by seasons per year (fig. 12). Seasonality factor is used to forecast and justify preventive measures in order to mitigate negative consequences. In winter, the frequency of emergencies increases by 30% compared with the frequency of events in spring, summer or autumn.

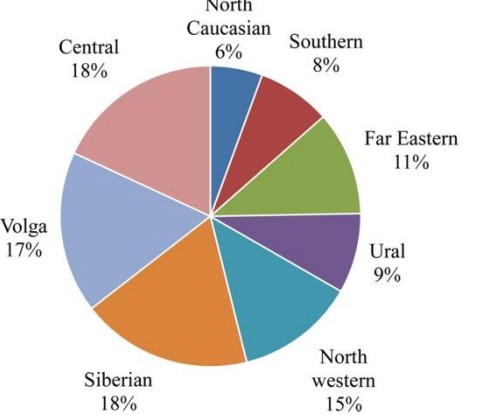

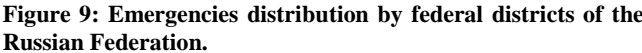

**Figure 9: Emergencies distribution by federal districts of the Russian Federation.**

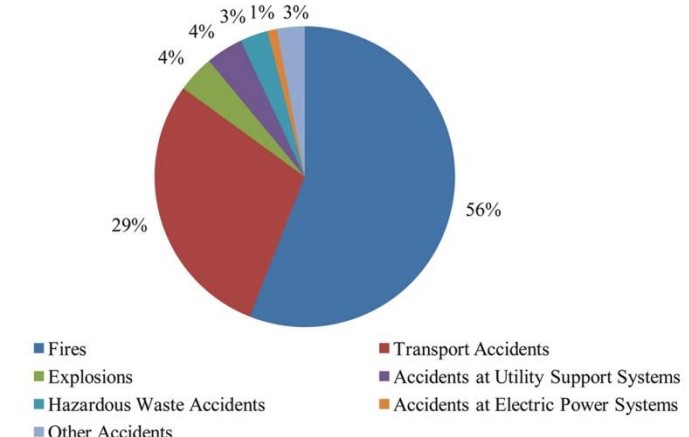

**Figure 10: Distribution of frequently repeated technological emergencies.**



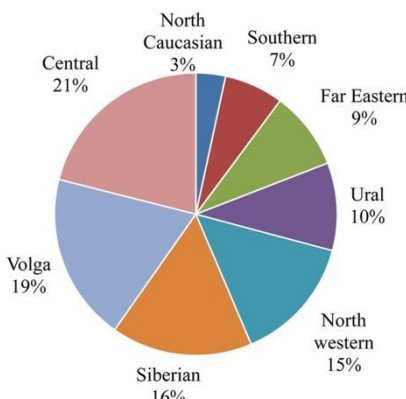
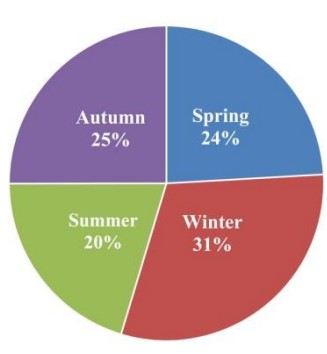

**Figure 11: Distribution of technological emergencies by federal districts of the Russian Federation.**   **Figure 12: Seasonal distribution of technological emergencies.**

Mapping the results of statistical analysis of EMERCOM DB data for the period from 1992 to 2018 (fig. 13) allows you to rank the subjects according to the frequency of occurrence of emergencies and select subjects for a more detailed assessment
or to check of the readiness of forces and resources to respond to emergencies and to implement preventive measures.

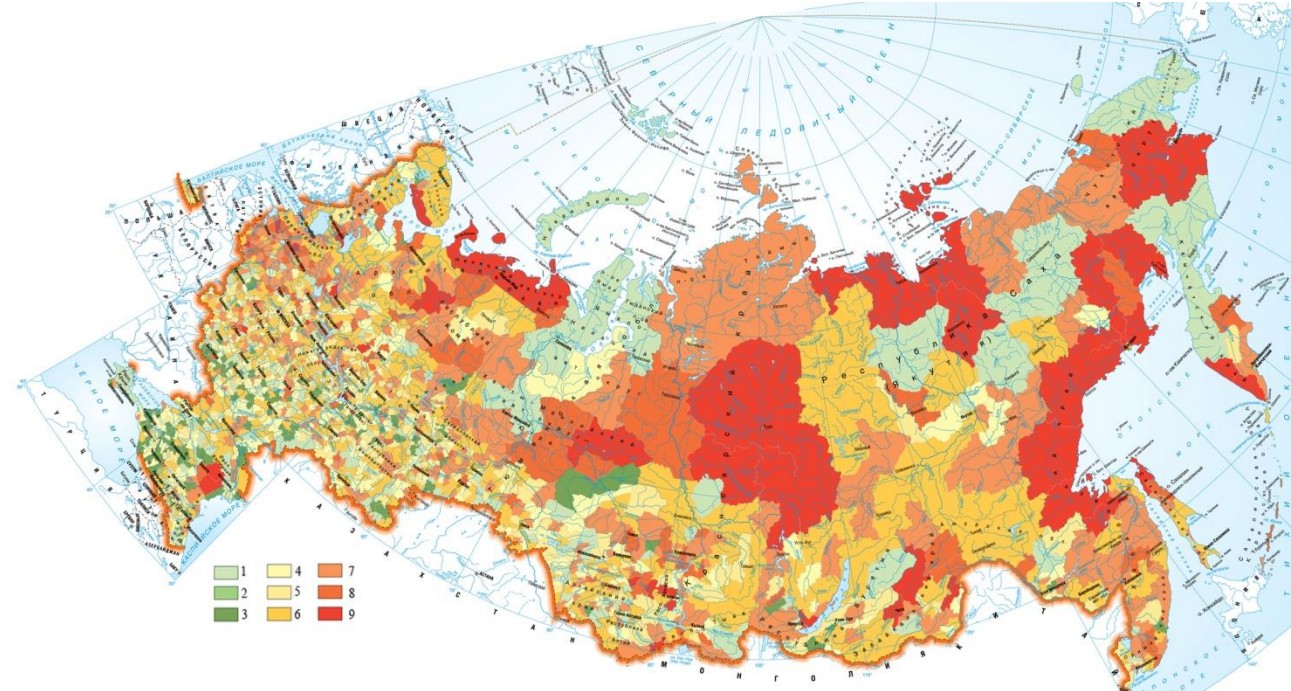

**Figure 13: Frequency of natural and technological emergencies according to statistical analysis of EMERCOM DB on events' information about emergency consequences for the period 1992-2008: number of events per year: 1 - less than 1; 2 – 1 up to 10; 3 – 10 up to 50; 4 – 50 up to 100; 5 – 100 up to 150; 6 – 150 up to 300; 7- 300 up to 600; 8- 600 up to 1,000; 9 – more than 1,000.**



The statistical data available in the EMERCOM DB and simulation models calibrated on its basis make it possible to update the information on individual types of emergencies. Figure 14 presents an updating example for individual risk assessments due to natural hazards where we incorporate six natural disasters, viz., earthquakes, floods, storms, landslides, mudflows and

avalanches.

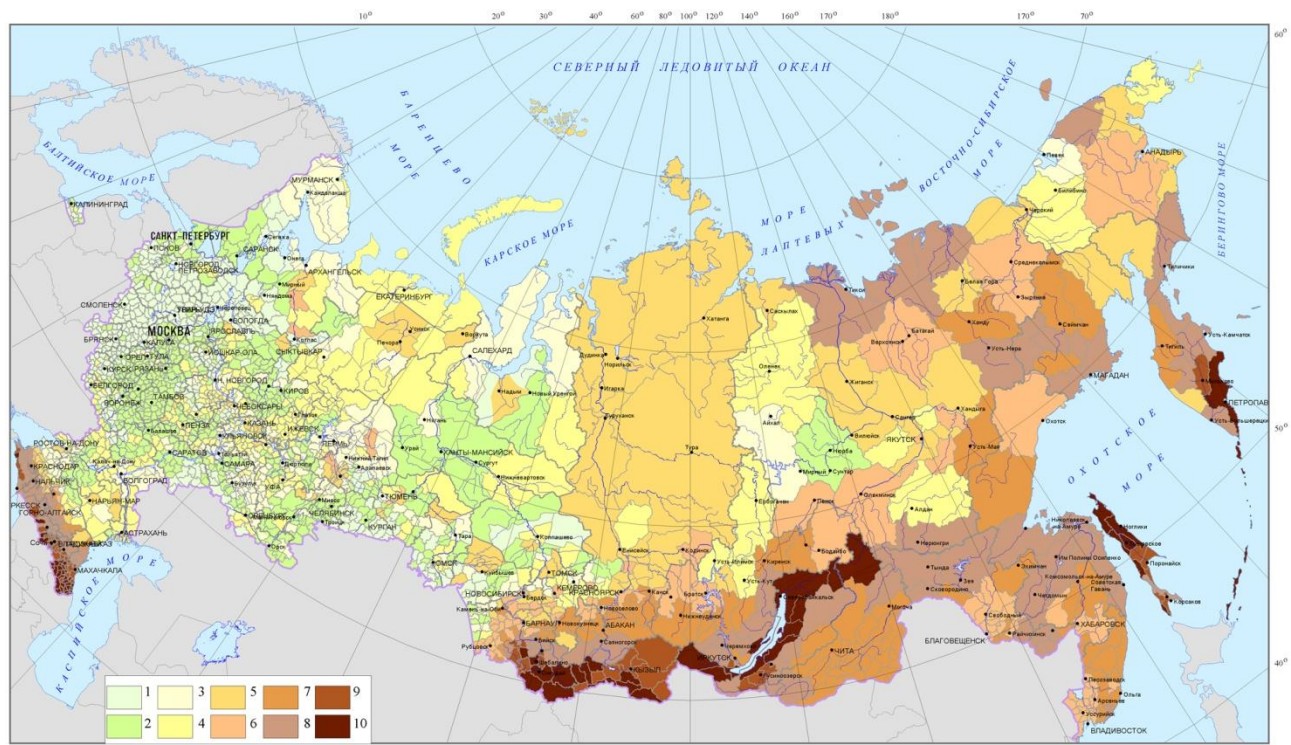

**Figure 14: Individual natural risk $R_3$ for the population of the Russian Federation: risk $R_s2$, $10^{-5}$ 1/year categories: 1 - less than 0.1; 2- 0.1 up to 1; 3 – 1 up to 2; 4 – 2 up to 5; 5 – 5 up to 10; 6 – 10 up to 15; 7 – 15 up to 30; 8 – 30 up to 100; 9 – 100 up to 150; 10 - more than 150.**

In our estimation of the parameters that represent an integrated risk caused by several natural disasters, we used probabilistic considerations. The probabilistic approach is in place here because an emergency situation that can occur for people is

evidently stochastic. We are not in a position to determine the intensity of the damaging factor with 100% certainty in the zone where the elements at risk are. The effects of similar factors on same-type elements at risk will yield different probabilities of damage to these elements. To calculate the risk parameters, we assumed the events to be independent, i.e., as concerns the social losses due to emergencies arising from natural disasters. We incorporated the probability for people to happen to be in the affected area, equally as the density of population in the larger cities with more than 500 thousand

inhabitants.





Our chief sources of data were thematic maps in vector form supplied with descriptions of the areas where various natural disasters can occur, and the associated parameters of the disasters are different. Statistical selections with descriptions of the locations and the extent of the damage due to these hazards as found in the EMERCOM DB for events and other archival databases, including the database of the Sergeev Institute of Environmental Geoscience of RAS (Osipov et al., 2009; Osipov

et al., 2011; Osipov et al., 2017), were employed to determine vulnerability parameters due to floods, storms, landslides, mudflows, and avalanches. A set of review seismic zoning maps for the Russian Federation area (Set…, 1998) was used as the chief data source for the level of earthquake hazards. The information about the distribution of residents and buildings was based on results of an inventory (fig. 2).

Estimates of integrated risk parameters as found for population centers were averaged for administrative areas. We have

made three maps of individual integrated risk due to natural hazards for the population of the Russian Federation (Frolova et al., 2014; Frolova et al. 2017): $R_e1$ is the probability of fatalities caused by the processes; $R_e2$ is the probability of fatalities or injuries of various degrees; $R_e3$ is the probability of fatalities or injuries of various degrees, or material losses.

The highest values of individual risk due to natural disasters in the map $R_e3$ (fig. 14) that apply to fatalities or injuries of various degrees, or material losses in the case of the six natural processes dealt with here were found for Kamchatka, the

Altai Republic, the Krasnodar and Trans-Baikal regions, the Republics of Buryatiya and Tyva, Sakhalin Island, the Altai Krai, and Northern Ossetia.

## 5 Conclusions

The article defines the role and place of the DB containing descriptions of events and their impact in the EMERCOM IS. The ordered event data obtained by EMERCOM and various agencies as a result of field survey and by remote sensing

techniques is used in the IS to address all the main problems in mitigating the impact of natural and technological emergencies.

The application of EMERCOM DB for events to calibrate IS mathematical models allows us to enhance the reliability of results from simulations of possible emergency impact, which affect the efficiency of decisions on response measures and facilitates rescue operations.

The above statistical analysis of data contained in the EMERCOM DB associated with the place and time, enabled us to generate maps of emergency frequency and risk indicators, ensuring rational use of resources for preventive measures implementation aimed at risk reduction, as well as improving the adequacy of long-term forecasts for strategic investment planning.

It was noted that the activity concerned with the recording, classification, and description of natural and technological

emergencies is the key issue in attempts to enhance the effectiveness of the Government's activities aimed at ensuring public safety against natural disasters and technological accidents. Created for these purposes, the EMERCOM IS, supported by the efforts of the National Center for Crisis Management of EMERCOM, needs further development and integration into the



network of similar systems providing services to other countries and the scientific community to minimize the level of risk. Integration involves solving problems of standard description of events, searching ways to ensure access to the functions of entering new information and updating the existing one. In this regard, the article notes the positive experience of CODATA efforts, which addresses the problem of integrating and analyzing big data in order to reduce the risk of natural hazards.

**Special issue statement.**

This article is part of the special issue "Natural hazard impacts on technological systems and infrastructures". It is a result of the EGU General Assembly 2019, Vienna, Austria, 7–12 April 2019.

**Acknowledgements**

The research was fulfilled according to the state project No. AAAA-A19-119021190077-6. Authors thank their colleagues for continuing support, discussion of different steps of EMERCOM IS DB on events development and impact data base usage for Extremum system mathematical models calibration. Special thanks are extended to the staff of Extreme Situations Research Center for their contribution to GIS environment development and colleagues in Federal Geophysical Survey of RAS for fruitful cooperation. Authors wish to acknowledge Dr. A. Petrosyan for providing helpful comments on this study.

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
