# Peer review of "Impact databases application for natural and technological risk management"

_Natural Hazards and Earth System Sciences, 2019_

## Referee Comment (RC1) · Anonymous Referee #1 · 12 Sep 2019

Review criteria and assesment

1. Does the paper address relevant scientific and/or technical questions within the scope of NHESS?

Yes, the paper covered the scientific topics of NHEES.

2. Does the paper present new data and/or novel concepts, ideas, tools, methods or results?

The paper presents a new concept and results, related to the data bases and applications to the risk management.

3. Are these up to international standards?

[Figure]

The international standards are rather wide in this direction, so the paper is included to them.

4. Are the scientific methods and assumptions valid and outlined clearly?

Yes, all scientific methods and assumptions are valid and presented clearly, by text, figures, tables and results.

5. Are the results sufficient to support the interpretations and the conclusions?

All results and conclusions support the interpretations and outcomes and are supported by several case-studies.

6. Does the author reach substantial conclusions?

The authors reach substantial conclusions in the scope of the paper related to the risc management of the natural and technological hazards.

7. Is the description of the data used, the methods used, the experiments and calculations made, and the results obtained sufficiently complete and accurate to allow their reproduction by fellow scientists (traceability of results)?

The traceability of results are sufficiently complete and accurate and allow the reproduction. This is supported by the description of the algorithm, data and methods of calculation.

8. Does the title clearly and unambiguously reflect the contents of the paper?

Yes, the title covers clearly the topic of the content.

9. Does the abstract provide a concise, complete and unambiguous summary of the work done and the results obtained?

Yes, Abstract provide correct summary of the work done.

10. Are the title and the abstract pertinent, and easy to understand to a wide and diversified audience?

The title and abstract are targeted to the professionals, but also are useful for the decision makers and everyday practicing people in the filed of risk management.

11. Are mathematical formulae, symbols, abbreviations and units correctly defined and used? If the formulae, symbols or abbreviations are numerous, are there tables or appendixes listing them?

The mathematical description is one of the strongest sides of the paper.

12. Is the size, quality and readability of each figure adequate to the type and quantity of data presented?

The size and quality of figures is representative to all data presented.

13. Does the author give proper credit to previous and/or related work, and does he/she indicate clearly his/her own contribution?

Yes, it is. Some additional references are possible to include. (for examp.Ranguelov B., 2011. Natural Hazards – nonlinearities and assessment., Acad. Publ. House (BAS), ISBN 978-954-332-419-7, 327 pp.)

14. Are the number and quality of the references appropriate?

Mostly the references are cited appropriately.

15. Are the references accessible by fellow scientists?

Most of the references are accessible to the scientific community.

16. Is the overall presentation well structured, clear and easy to understand by a wide and general audience?

Yes, especially by the people working in the field of the hazard assessment and risk management .

17. Is the length of the paper adequate, too long or too short?

The length is normal.

18. Is there any part of the paper (title, abstract, main text, formulae, symbols, figures and their captions, tables, list of references, appendixes) that needs to be clarified, reduced, added, combined, or eliminated?

In the list of References could be added some publications which is possible to be used for extended research (for examp. B. Ranguelov., A. Frantsova., 2017. Multi-hazards early warnings. Research, models and Bulgarian expertise., LAMBERT Academic Publishing., Saarbrucken, 224 pp. ISBN: 978-620-2-07727-9 and Paskaleva I., B.Ranguelov.2015. Lessons learned by recently happened natural disasters and future research needs., In "Engaging the Public to Fight the Consequences of Terrorism and Disasters." Eds. I. Apostol, J.Mamasaklihsi, D.Subotta, D. Reimer. NATO Sci. for Peace and Security, E: Human and Societal Dynamics, vol. 120., IOS Press., 257-268 pp.)

19. Is the technical language precise and understandable by fellow scientists?

The technical and scientific language is understandable by the specialized auditory.

20. Is the English language of good quality, fluent, simple and easy to read and understand by a wide and diversified audience?

I'm not a professional linguist so it is difficult to me to asses the quality of English language used.

Evaluation synthesis

The paper entirely covered the scientific topics of NHEES. The paper presents a new concept and results, related to the data bases and applications to the risk management.The international standards are rather wide in this direction, so the paper is included to them, but all scientific methods and assumptions are valid and presented clearly, by text, figures, tables and results. All results and conclusions support the interpretations and outcomes and are supported by several case-studies.

The authors reach substantial conclusions in the scope of the paper related to the risk management of the natural and technological hazards. The traceability of results are sufficiently complete and accurate and allow the reproduction. This is supported by the description of the algorithm, data and methods of calculation.

The title of the paper covers clearly the topic of the content and the abstract provides correct summary of the work done. The title and abstract are targeted to the professionals, but also are useful for the decision makers and everyday practicing people in the filed of risk management.

The mathematical description is one of the strongest sides of the paper. The statistics provided are clear, correct and supported by several figures and schemes. The size and quality of figures is representative to all data used and interpreted.

The reference list is dominated by Russian publications. Some additional references are possible to be included. (for examp. Ranguelov B., 2011. Natural Hazards – nonlinearities and assessment., Acad. Publ. House (BAS), ISBN 978-954-332-419-7, 327 pp.) to the section where the nonlinear effects are described.

In general mostly the references are cited appropriately and most of the references are accessible to the scientific community.

The presentation of the paper is well structured, clear and easy to understand by a wide and general audience and especially by the people working in the field of the hazard assessment and risk management .The length is normal.

In the list of References could be added some publications which is possible to be used for extended research (for examp. B. Ranguelov., A. Frantsova., 2017. Multi-hazards early warnings. Research, models and Bulgarian expertise., LAMBERT Academic Publishing., Saarbrucken, 224 pp. ISBN: 978-620-2-07727-9 and Paskaleva I., B.Ranguelov.2015. Lessons learned by recently happened natural disasters and future research needs., In "Engaging the Public to Fight the Consequences of Terrorism

and Disasters." Eds. I. Apostol, J.Mamasaklihsi, D.Subotta, D. Reimer. NATO Sci. for Peace and Security, E: Human and Societal Dynamics, vol. 120., IOS Press., 257-268 pp.)

The technical and scientific language is understandable by the specialized auditory. I'm not a professional linguist so it is difficult to me to asses the quality of English language used.

In the list of References have to be added some publications which is possible to be used for extended research on the topic presented by the paper (for examp. B. Ranguelov., A. Frantsova., 2017. Multihazards early warnings. Research, models and Bulgarian expertise., LAMBERT Academic Publishing., Saarbrucken, 224 pp. ISBN: 978-620-2-07727-9

Paskaleva I., B.Ranguelov.2015. Lessons learned by recently happened natural disasters and future research needs., In "Engaging the Public to Fight the Consequences of Terrorism and Disasters." Eds. I. Apostol, J.Mamasaklihsi, D.Subotta, D. Reimer. NATO Sci. for Peace and Security, E: Human and Societal Dynamics, vol. 120., IOS Press., 257-268 pp.)

Ranguelov B., 2011. Natural Hazards – nonlinearities and assessment., Acad. Publ. House (BAS), ISBN 978-954-332-419-7, 327 pp.

---

## Short Comment (SC1) · 13 Sep 2019

We are grateful to the Referee #1 for interest to our study and positive estimation of our results delivered in discussed paper. It will be our pleasure to analyze the recent results published in the papers recommended by Referee, use them for extended research and add to reference list of thr discussed paper.

---

## Referee Comment (RC2) · Anonymous Referee #2 · 19 Sep 2019

Manuscript number: NHESS-2019-264 https://doi.org/10.5194/nhess-2019-264 Preprint. Discussion started: 20 August 2019

Title: Impact data bases application for natural and technological risk management General comments I reviewed the article titled " Impact data bases application for natural and technological risk management" and I observed that the manuscript investigates an important concern of database for disaster risk assessment and management. In addition, an application on the usage of impact data for calibration of earthquake loss models in order to increase the reliability of near real-time estimates is enlightening. In my opinion, such a study is of great significance, given the growing use of self-learning computer and artificial intelligence which could play (or is playing) a major role in altering the face of hazard and risk analysis. The paper entirely covers the scientific topics

of Natural Hazard and Earth Systems Sciences journal and special issue "Natural hazard impacts on technological systems and infrastructures". I am convinced with the -of-the-art, methodology results and conclusions. Overall the manuscript is well written, but it would be great if the author (s) could take care of the following concerns into their final version.

Specific comments I had a bit of difficulty in understanding the definition of impact databases? I would prefer to at least mention what the author means by impact databases? In the title, the author writes data bases, whereas in abstract databases (be consistent!!).

Overall, the research is of high quality and I want to congratulate the authors on an interesting and well-documented investigation.

---

## Author Comment (AC1) · 21 Sep 2019

We are grateful to the Referee #2 for the interest to our study and high estimation of our research quality. We will definitely be consistent in writing word "database" in the final version of the paper. Speaking about our understanding on "impact database", we consider that EMERCOM database about natural and technological events in Russia, which contains the information about event date, location, parameters, its social and economic consequences including information about casualties and damage to different elements at risk, as well as optionally information on emergency operations and used forces and resources, may be called as "impact database". To some extent the EMERCOM impact database on events of different origin is similar to Earthquake Impact Databases (http://www.ceqid.org/CEQID/Home.aspx;

https://earthquake-report.com/2018/02/10/earthquake-impact-database-2018).

Please also note the supplement to this comment:
https://www.nat-hazards-earth-syst-sci-discuss.net/nhess-2019-264/nhess-2019-264-AC1-supplement.pdf

---

## Author Comment (AC3) · 7 Nov 2019

Answer to the Editor comments Dear Maria,

Thank you for your comments and positive decision about our manuscript publication with minor revisions. The infrastructure described in the manuscript was developed within the Russian Federal Programs and is maintained by EMERCOM National Center for Crisis Management. We suppose that similar infrastructures exist in many countries, especially in National Emergency Management Agencies, and used for natural and technological emergency management at the country level. Usually such infrastructures are accessible within specific projects to solve definite tasks. The examples of these data usage within national projects are given in our paper. This and other

similar infrastructures should be definitely accessed within the international projects as well. We do our best to participate in international activity aimed at disaster risk reduction with special attention to earthquake risk management. In 2004 we were invited to JRC to share our experience on global system Extremum assigned for near real time loss assessment due to earthquakes. At that time Extremum and its latest versions were most probably the best probabilistic earthquake mortality model existing and by that time it had been tested and improved for over 10 years. It was a large interest on both sides, JRC and Russian Academy of Sciences, to collaborate on the issue of near real-time earthquake impact modelling and alerting. Both parties were aware that a single model is not enough for informed decisions. Decision makers wanted to be able to compare and/or combine the outcomes of many models, many approaches and many methodologies. We are quite aware that impact databases on earthquake consequences are very important in order to increase the reliability of loss estimations by application of existing near real time global Systems. In 2010 my coauthor and me prepared the proposal for the 27th CODATA General Assembly about new Task Group "Knowledge-Base on Physical and Socio-Economical Consequences of Damaging Earthquakes". Many scientists who were ready to take part in this TG activity are still involved in this field (see attached file). For instance, in 2011-2014 prof. Robin Spence was one of the scientific leaders of the GEM Project aimed at development of the Global Earthquake Consequence Database. In 2018 we initiated together with him and CODATA the ESC special session in order to investigate the state of the art of existing impact databases on past earthquake consequences, to summarize different factors affecting on the reliability of near real time loss estimations and identify the way to minimize their influence. The important aim of this session was to explore ways in which the datasets on earthquake consequences from different countries and institutions could possibly be merged, and to what extent these data should be harmonized, as well as to discuss different issues dealt with creation of distributed data base. At present we continue this activity within the CODATA Task Group on Linked Open Data for Global Disaster Risk Research. This September we issued the study

report "Next Generation Disaster Data Infrastructure" (see attached file). In this Report the proper attention is given to impact databases on earthquakes and loss estimations in emergency mode. The general issues of disaster data collection and transmission, standards and formats, quality and control, availability and others are discussed. We do hope that the proposed in the Report the next generation of disaster data infrastructure, which includes both novel and the most essential information systems and services that a country or a region can depend on to successfully gather, process and display disaster data to reduce the impact of natural hazards, including earthquakes. Therefore, in our manuscript we gave reference to this study report and do hope it will stimulate the development of distributed databases on separate disasters in order to include the data from national database similar to ours described in the manuscript. I am sorry for this long response on your comments on cooperation programmes. I am afraid that we will need another paper in order to analyze the achievements on development impact databases and increasing loss estimation reliability within international cooperation activity. In the manuscript submitted for your special issue we did our best to describe Russian national IS and its impact database, as well as the results of our study of these data usage within the Russian national projects. In the final version of the manuscript we took into account your and Referee #1 comments and added new references of foreign investigators.
* * *
Dear Maria,

Thank you for your comments and positive decision about our manuscript publication with minor revisions.

The infrastructure described in the manuscript was developed within the Russian Federal Programs and is maintained by EMERCOM National Center for Crisis Management. We suppose that similar infrastructures exist in many countries, especially in National Emergency Management Agencies, and used for natural and technological emergency management at the country level. Usually such infrastructures are accessible within specific projects to solve definite tasks. The examples of these data usage within national projects are given in our paper. This and other similar infrastructures should be definitely accessed within the international projects as well. We do our best to participate in international activity aimed at disaster risk reduction with special attention to earthquake risk management.

In 2004 we were invited to JRC to share our experience on global system Extremum assigned for near real time loss assessment due to earthquakes. At that time Extremum and its latest versions were most probably the best probabilistic earthquake mortality model existing and by that time it had been tested and improved for over 10 years. It was a large interest on both sides, JRC and Russian Academy of Sciences, to collaborate on the issue of near real-time earthquake impact modelling and alerting. Both parties were aware that a single model is not enough for informed decisions. Decision makers wanted to be able to compare and/or combine the outcomes of many models, many approaches and many methodologies.

We are quite aware that impact databases on earthquake consequences are very important in order to increase the reliability of loss estimations by application of existing near real time global Systems. In 2010 my coauthor and me prepared the proposal for the 27th CODATA General Assembly about new Task Group "Knowledge-Base on Physical and Socio-Economical Consequences of Damaging Earthquakes". Many scientists who were ready to take part in this TG activity are still involved in this field (see attached file).

For instance, in 2011-2014 prof. Robin Spence was one of the scientific leaders of the GEM Project aimed at development of the Global Earthquake Consequence Database. In 2018 we initiated together with him and CODATA the ESC special session in order to investigate the state of the art of existing impact databases on past earthquake consequences, to summarize different factors affecting on the reliability of near real time loss estimations and identify the way to minimize their influence. The important aim of this session was to explore ways in which the datasets on earthquake consequences from different countries and institutions could possibly be merged, and to what extent these data should be harmonized, as well as to discuss different issues dealt with creation of distributed data base.

At present we continue this activity within the CODATA Task Group on Linked Open Data for Global Disaster Risk Research. This September we issued the study report "Next Generation Disaster Data Infrastructure" (see attached file). In this Report the proper attention is given to impact databases on earthquakes and loss estimations in emergency mode. The general issues of disaster data collection and transmission, standards and formats, quality and control, availability and others are discussed. We do hope that the proposed in the Report the next generation of disaster data infrastructure, which includes both novel and the most essential information systems and services that a country or a region can depend on to successfully gather, process and display disaster data to reduce the impact of natural hazards, including earthquakes. Therefore, in our manuscript we gave reference to this study report and do hope it will stimulate the development of distributed databases on separate disasters in order to include the data from national database similar to ours described in the manuscript.

I am sorry for this long response on your comments on cooperation programmes. I am afraid that we will need another paper in order to analyze the achievements on development impact databases and increasing loss estimation reliability within international cooperation activity. In the manuscript submitted for your special issue we did our best to describe Russian national IS and its impact database, as well as the results of our study of these data usage within the Russian national projects.

In the final version of the manuscript we took into account your and Referee #1 comments and added new references of foreign investigators.

**Fig. 1.**

**NEW TASK GROUP PROPOSAL FOR PRESENTATION TO THE
27th CODATA GENERAL ASSEMBLY
Cape Town, 28-29th October 2010**

**1 Name of the Proposed Task Group**

**Knowledge-Base on Physical and Socio-Economical Consequences of
Damaging Earthquakes**

**2 Objective(s) of the Proposed Task Group**

Social and economic losses caused by strong earthquakes increase markedly in the recent
decades, which is a definite trend of Society's evolution. By far, the best way to
mitigating the effects of an earthquake on population and artifacts, is to apply prevention
measures before the earthquake occurs : as earthquake occurrence is not predictable in
the present state of knowledge, one is led to implement prevention measures relying on
an assessment of what could be the consequences of an earthquake likely to occur in an
earthquake-prone region; in other words to imagine a scenario of the occurrence of the
given earthquake in the region under investigation. One can also apply the scenario
approach in an emergency mode just upon occurrence of the event; indeed, it is then too
late to think of prevention measures, but the outcome of this fast approach can then be
hopefully useful for advising, as early as feasible, on how to best orient the rescue teams.
Both approaches require ideally a good background knowledge of the objects-at-risk and
the way they respond to potential earthquake solicitation. Simulation codes exist, both in
emergency mode and in longer-term mode; the quality of the assessment output is
directly dependent upon the quality of input data and of the simulation models. An
obvious way of improving the whole process is to confront the assessment output with
the known consequences of previous events; *i.e*. data on impact of past earthquakes could
help "calibrating" somehow the simulation models; furthermore, scenario earthquake
approaches suffer from more or less badly-known parameters (inventory of objects-at-
risk, vulnerability/fragility functions of buildings submitted to shaking, *etc*.) : to a certain
extent, these weaknesses can be partially mended through calibration procedure, in
addition to improvement of available databases. In this respect, the information on
physical and socio-economical consequences of past damaging earthquakes is very
critical.

At the moment, data sets on impact of past earthquakes are not readily accessible to many
potential users; actually, if partial data sets have been developed here and there, no
significant initiative has yet been taken to collect, organize, and make easily available the
corresponding data

The role of the foreseen Task Group is to prepare the construction and initial
development (structure and content) of a database on earthquake impact data. Its role will

**Fig. 2.**

[Figure]

**Fig. 3.**

---

## Editor Comment (EC1) · Maria Bostenaru Dan (Editor) · 8 Nov 2019

Many thanks for your thorough and detailed response! It is indeed very interesting to read about all these initiatives. It would be very good if reference is included to those attached to your response, and sufficient. Looking forward for a future article or presentation describing what is in the response!

---

## Author Comment (AC4) · 8 Nov 2019

Authors response to the editor comments

Dear Maria,

Thanks again for your kind comments. According to your comment we added the reference for the CODATA study report to the list of references. In the previous version of the manuscript it was only mentioned as web reference https://doi.org/10.5281/zenodo.3406127 on line #60. We are also looking for future cooperation.
* * *
2019-264, 2019.

Interactive comment on "Impact databases application for natural and technological risk management"
by Nina I. Frolova et al.
Maria Bostenaru Dan (Editor)
csipike@web.de

Many thanks for your thorough and detailed response! It is indeed very interesting to read about all these initiatives. It would be very good if reference is included to those attached to your response, and sufficient. Looking forward for a future article or presentation describing what is in the response!

Authors response to the editor comments

Dear Maria,

Thanks again for your kind comments.
According to your comment we added the reference for the CODATA study report to the list of references. In the previous version of the manuscript it was only mentioned as web reference https://doi.org/10.5281/zenodo.3406127 on line #60.
We are also looking for future cooperation.

**Fig. 1.**
**Impact databases application for natural and technological risk management**

Nina I. Frolova[1], Valery I. Larionov[1], Jean Bonnin[2], Sergey P. Suchshev[3], Alexander Ugarov[3], Nataliya Malaeva[3]

[1] Seismological Center of IGE, Russian Academy of Sciences, Moscow 101000, Russia
[2] Institut de Physique du Globe, University of Strasbourg, Strasbourg F-67084, France
[3] Extreme Situations Research Center, Moscow 127015, Russia

*Correspondence to*: Nina I. Frolova (frolova@esrc.ru)

**Abstract.** Impact databases development and application for risk analysis and management promotes the usage of self-learning computer systems with elements of artificial intelligence. Such systems learning could be successful when the databases store the complete information about each event, parameters of the simulation models, the range of its application and residual errors. Each new description included in the database could increase the reliability of the results obtained with application of simulation models. The calibration of mathematical models is the first step to self-learning of automated systems. The article describes the events' database structure, and examples of calibrated computer models as applied to the impact of expected emergencies and risk indicators assessment. Examples of database statistics usage in order to rank the subjects of the Russian Federation by the frequency of emergencies of different character, as well as risk indicators are given.

**1 Introduction**

Analysis of the natural and technological emergencies consequences gives an evidence that natural hazards and technological disasters pose an increasing threat to the safety of citizens and the economy of the Russian Federation. The increasing severity of the impact indicates the need to improve the effectiveness of measures aimed at risk reduction. The Ministry of Emergency Situations (EMERCOM) of the Russian Federation considers preventive measures as the priority. They are based on application of information systems (IS) that provide reliable forecasts, including a reliable assessment of a spatially distributed indicator that characterizes the safety of a region, which makes it possible to rationally distribute forces and resources in order to reduce risk.

Another important task, which may be solved with IS usage, is to enhance the efficiency of rescue operations. This could be achieved by higher reliability of the operational forecast of situations based on the data contained in the database of events (DB). Examples of successful rescue operations accumulated in DB facilitate the decision-making process and reduce the time when people stay in the affected area, which results in decreasing the fatality likelihood.

[Figure]

**Fig. 2.**

---

## Author Response (AR1)

[revised manuscript text omitted]

310   LODGD 2019, Next Generation Disaster Data Infrastructure -White Paper. CODATA Task Group, Linked Open Data for Global Disaster Risk Research (LODGD). September 2019. Paris. DOI 10.5281/zenodo.3406127.

Osipov, V.I., Burova, V.N., Pyrchenko, V.A., Petrasov, A.V.: The database of natural hazards as the basis for the study of their development regularities and consequences prediction, in: Extreme natural phenomena and disasters, 2 volumes. Volume 1: Assessments and ways of mitigating negative consequences of extreme natural phenomena, Probel 2000 Publ.,

315   Moscow, Russian Federation, pp. 436-445, 2010. (in Russian)

Osipov, V.I., Frolova, N.I., Sushchev, S.P., and Larionov, V.I.: Assessment of seismic and natural risk for population and territory of the Russian Federation, in: Extreme Natural Phenomena and Disasters, vol, 2: PROBEL 2000 Publ., Moscow, Russian Federation, pp.28-48., 2011. (in Russian)

Osipov, V.I., Larionov, V. I., Burova, V.N., Frolova, N.I., and Sushchev, S. P.: Methodology of natural risk assessment in

320   Russia, Nat. Hazards, vol. 88, Is. 1, pp. 17-41. DOI 10.1007/s11069-017-2780-z, 2017.

Paskaleva, I., Ranguelov, B.: Lessons learned by recently happened natural disasters and future research needs, in "Engaging the Public to Fight the Consequences of Terrorism and Disasters." Eds. I. Apostol, J.Mamasaklihsi, D.Subotta, D. Reimer. NATO Sci. for Peace and Security, E: Human and Societal Dynamics, vol. 120., IOS Press., pp. 257-268, 2015.

Ranguelov, B.: Natural Hazards – nonlinearities and assessment., Acad. Publ. House (BAS), ISBN 978-954-332-419-7, 327 pp., 2011.

325

Ranguelov, B., Frantsova, A.: Multihazards early warnings. Research, models and Bulgarian expertise., LAMBERT Academic Publishing, Saarbrucken, 224 pp. ISBN: 978-620-2-07727-9, 2017.

Rogozhin, E.A., Lutikov, A.I., and Ovcuchenko A.N.: The Detailed Assessment of the Seismic Risk in the Territory of North Caucasus, Earthquake Engineering Constructions Safety, no.5, pp. 14-19, 2013. (in Russian) http://www.seismic-safety.ru/

330

Rovida, A., Locati, M., Camassi, R., Lolli, B., and Gasperini, P.: CPTI15, the 2015 version of the Parametric Catalogue of Italian Earthquakes, Istituto Nazionale di Geofisica e Vulcanologia, http://doi.org/10.6092/INGV.IT-CPTI15, 2016.

Set of review seismic zoning maps OSR-97 A, B, C and other materials for Constructions standards and rules. 1998 - SNiP "Construction in earthquake prone areas". Moscow: Joint Institute of Physics of the Earth, 1998. (in Russian)

Shebalin, N.V.: Reference earthquakes and macroseismic field equations, in: New catalogue of strong earthquakes for the

335

USSR territory from ancient times till 1975, Nauka, Moscow, Russian Federation, pp. 20–30. 1977. (in Russian)

Shebalin, N.V., Ershov, I.A., Gekhman, A.S., and Shestoperov, G.S.: Development of improved version of seismic intensity scale (MMSK-86) on the bases of MSK-64 scale and scale of Joint Council on Seismology and Earthquake Engineering – 73. Report on scientific research study within the Federal Program 0.74.03, number of state registration 01814003271, Joint Council on Seismology and Earthquake Engineering, Academy of Sciences of the USSR, Moscow, Russian Federation.

340

1986. (in Russian)

So, E.: Introduction to the GEM Earthquake Consequences Database (GEMECD), GEM Technical Report, 2014-14. V1.0.0. 158 P., GEM Foundation, Pavia, Italy. DOI: 10.13117/ GEM.VULN-MOD.TR2014.14, 2014.

Spence, R., So, E., Jenkins, S., Coburn, A., and Ruffle, S.: A Global Earthquake Building Damage and Casualty Database, in: Spence, R., So, E., and Scawthorn, C. (eds): Human Casualties in Earthquakes. Advances in Natural and

345

Technological Hazards Research, vol 29. Springer, Dordrecht doi.org/10.1007/978-90-481-9455-1_5, pp. 65-79, 2011.

Varazanashvili, O., Tsereteli, N., Bonali F.L. et al. GeoInt: the first macroseismic intensity database for the Republic of Georgia, Journal of Seismology, vol. 22, Is. 4, pp. 625-667. DOI: htts://doi.org/10.1007/s10950-017-9726-5, 2018.

**LIST OF RESPONSES TO THE REVIEWERS COMMENTS**
on "Impact databases application for natural and technological risk management"

**Anonymous Referee #1**

COMMENTS FOR THE AUTHORS:
13. Does the author give proper credit to previous and/or related work, and does he/she indicate clearly his/her own contribution?
Yes, it is. Some additional references are possible to include. (for examp.Ranguelov B., 2011. Natural Hazards – nonlinearities and assessment., Acad. Publ. House (BAS), ISBN 978-954-332-419-7, 327 pp.)

18. Is there any part of the paper (title, abstract, main text, formulae, symbols, figures and their captions, tables, list of references, appendixes) that needs to be clarified, reduced, added, combined, or eliminated?
In the list of References could be added some publications which is possible to be used for extended research (for examp. B. Ranguelov., A. Frantsova., 2017. Multihazards early warnings. Research, models and Bulgarian expertise., LAMBERT Academic Publishing., Saarbrucken, 224 pp. ISBN: 978-620-2-07727-9 and Paskaleva I., B.Ranguelov.2015. Lessons learned by recently happened natural disasters and future research needs., In "Engaging the Public to Fight the Consequences of Terrorism and Disasters." Eds. I. Apostol, J.Mamasaklihsi, D.Subotta, D. Reimer. NATO Sci. for Peace and Security, E: Human and Societal Dynamics, vol. 120., IOS Press., 257-268 pp.)

Evaluation synthesis
The reference list is dominated by Russian publications. Some additional references are possible to be included. (for examp. Ranguelov B., 2011. Natural Hazards – nonlinearities and assessment., Acad. Publ. House (BAS), ISBN 978-954-332-419-7, 327 pp.) to the section where the nonlinear effects are described.

In the list of References could be added some publications which is possible to be used for extended research (for examp. B. Ranguelov., A. Frantsova., 2017. Multihazards early warnings. Research, models and Bulgarian expertise., LAMBERT Academic Publishing., Saarbrucken, 224 pp. ISBN: 978-620-2-07727-9 and Paskaleva I., B.Ranguelov.2015. Lessons learned by recently happened natural disasters and future research needs., In "Engaging the Public to Fight the Consequences of Terrorism and Disasters." Eds. I. Apostol, J.Mamasaklihsi, D.Subotta, D. Reimer. NATO Sci. for Peace and Security, E: Human and Societal Dynamics, vol. 120., IOS Press., 257-268 pp.)

In the list of References have to be added some publications which is possible to be used for extended research on the topic presented by the paper (for examp. B. Ranguelov., A. Frantsova., 2017. Multihazards early warnings. Research, models and Bulgarian expertise., LAMBERT Academic Publishing., Saarbrucken, 224 pp. ISBN: 978-620-2-07727-9

Paskaleva I., B.Ranguelov.2015. Lessons learned by recently happened natural disasters and future research needs., In "Engaging the Public to Fight the Consequences of Terrorism and Disasters." Eds. I. Apostol, J.Mamasaklihsi, D.Subotta, D. Reimer. NATO Sci. for Peace and Security, E: Human and Societal Dynamics, vol. 120., IOS Press., 257-268 pp.)

Ranguelov B., 2011. Natural Hazards – nonlinearities and assessment., Acad. Publ. House (BAS), ISBN 978-954-332-419-7, 327 pp.

*Answer*: The authors analyzed the recent results published in the papers recommended by Referee #1 and the corresponding references will be made in the final version of the manuscript.

**Anonymous Referee #2**

COMMENTS FOR THE AUTHORS:
Specific comments I had a bit of difficulty in understanding the definition of impact databases? I would prefer to at least mention what the author means by impact databases? In the title, the author writes data bases, whereas in abstract databases (be consistent!!).

*Answer*: The authors will be consistent in writing word "database" in the final version of the paper.
For better understanding of "impact database" the links to similar impact databases will be given in the final version of the manuscript.

Dear Maria,

Thank you for your comments and positive decision about our manuscript publication with minor revisions.

The infrastructure described in the manuscript was developed within the Russian Federal Programs and is maintained by EMERCOM National Center for Crisis Management. We suppose that similar infrastructures exist in many countries, especially in National Emergency Management Agencies, and used for natural and technological emergency management at the country level. Usually such infrastructures are accessible within specific projects to solve definite tasks. The examples of these data usage within national projects are given in our paper. This and other similar infrastructures should be definitely accessed within the international projects as well. We do our best to participate in international activity aimed at disaster risk reduction with special attention to earthquake risk management.

In 2004 we were invited to JRC to share our experience on global system Extremum assigned for near real time loss assessment due to earthquakes. At that time Extremum and its latest versions were most probably the best probabilistic earthquake mortality model existing and by that time it had been tested and improved for over 10 years. It was a large interest on both sides, JRC and Russian Academy of Sciences, to collaborate on the issue of near real-time earthquake impact modelling and alerting. Both parties were aware that a single model is not enough for informed decisions. Decision makers wanted to be able to compare and/or combine the outcomes of many models, many approaches and many methodologies.

We are quite aware that impact databases on earthquake consequences are very important in order to increase the reliability of loss estimations by application of existing near real time global Systems. In 2010 my coauthor and me prepared the proposal for the 27th CODATA General Assembly about new Task Group "Knowledge-Base on Physical and Socio-Economical Consequences of Damaging Earthquakes". Many scientists who were ready to take part in this TG activity are still involved in this field (see attached file).

For instance, in 2011-2014 prof. Robin Spence was one of the scientific leaders of the GEM Project aimed at development of the Global Earthquake Consequence Database. In 2018 we initiated together with him and CODATA the ESC special session in order to investigate the state of the art of existing impact databases on past earthquake consequences, to summarize different factors affecting on the reliability of near real time loss estimations and identify the way to minimize their influence. The important aim of this session was to explore ways in which the datasets on earthquake consequences from different countries and institutions could possibly be merged, and to what extent these data should be harmonized, as well as to discuss different issues dealt with creation of distributed data base.

At present we continue this activity within the CODATA Task Group on Linked Open Data for Global Disaster Risk Research. This September we issued the study report "Next Generation Disaster Data Infrastructure" (see attached file). In this Report the proper attention is given to impact databases on earthquakes and loss estimations in emergency mode. The general issues of disaster data collection and transmission, standards and formats, quality and control, availability and others are discussed. We do hope that the proposed in the Report the next generation of disaster data infrastructure, which includes both novel and the most essential information systems and services that a country or a region can depend on to successfully gather, process and display disaster data to reduce the impact of natural hazards, including earthquakes. Therefore, in our manuscript we gave reference to this study report and do hope it will stimulate the development of distributed databases on separate disasters in order to include the data from national database similar to ours described in the manuscript.

I am sorry for this long response on your comments on cooperation programmes. I am afraid that we will need another paper in order to analyze the achievements on development impact databases and increasing loss estimation reliability within international cooperation activity. In the manuscript submitted for your special issue we did our best to describe Russian national IS and its impact database, as well as the results of our study of these data usage within the Russian national projects.

In the final version of the manuscript we took into account your and Referee #1 comments and added new references of foreign investigators.

Interactive comment on "Impact databases application for natural and technological risk management"
by Nina I. Frolova et al.
Maria Bostenaru Dan (Editor)
csipike@web.de

Many thanks for your thorough and detailed response! It is indeed very interesting to read about all these initiatives. It would be very good if reference is included to those attached to your response, and sufficient. Looking forward for a future article or presentation describing what is in the response!

Authors response to the editor comments

Dear Maria,

Thanks again for your kind comments.
According to your comment we added the reference for the CODATA study report to the list of references. In the previous version of the manuscript it was only mentioned as web reference https://doi.org/10.5281/zenodo.3406127 on line #60.
We are also looking for future cooperation.